# Directed propulsion of spherical particles along three dimensional helical trajectories

Jin Gyun Lee[1], Ada M. Brooks[2], William A. Shelton[1], Kyle J. M. Bishop [3] & Bhuvnesh Bharti [1]

Active colloids are a class of microparticles that 'swim' through fluids by breaking the symmetry of the force distribution on their surfaces. Our ability to direct these particles along complex trajectories in three-dimensional (3D) space requires strategies to encode the desired forces and torques at the single particle level. Here, we show that spherical colloids with metal patches of low symmetry self-propel along non-linear 3D trajectories when powered remotely by an alternating current (AC) electric field. In particular, particles with triangular patches of approximate mirror symmetry trace helical paths along the axis of the field. We demonstrate that the speed and shape of the particle's trajectory can be tuned by the applied field strength and the patch geometry. We show that helical motion can enhance particle transport through porous materials with implications for the design of microrobots that can navigate complex environments.

[1] Cain Department of Chemical Engineering, Louisiana State University, Baton Rouge, LA 70803, USA. [2] Department of Chemical Engineering, Pennsylvania State University, University Park, PA 16802, USA. [3] Department of Chemical Engineering, Columbia University, New York, NY 10027, USA. Correspondence and requests for materials should be addressed to B.B. (email: bbharti@lsu.edu)

Helical motion is observed at all length scales, from the trajectory of subatomic particles in magnetic fields to the migration of interplanetary bodies in space[1–3]. At low Reynolds numbers, microorganisms and spermatozoa self-propel along helical paths by non-reciprocal motions of their chiral components such as flagella[4,5]. In these biological swimmers, helical motion facilitates chemotaxis[6] whereby the rotational component enables scanning of three-dimensional (3D) space to determine the direction of an external stimulus, while the translational component drives migration toward the stimulus. The helical motions of flagellated microorganisms are known to enhance their motility[4] and facilitate navigation through complex environments[7]. However, encoding similar helical motions in engineered colloids remains a challenge, largely due to limitations in synthesizing particles with the necessary symmetry, shape, and/or composition[8,9].

Active colloids convert energy from their environment (e.g., chemical fuels, external fields) into fluid flows that propel particle motions directed by asymmetries in particle shape and composition[9–17]. For micron-scale particles, gravitational forces are often significant thereby confining the motions of self-propelled colloids to the two-dimensional environment above a planar substrate. By contrast, the propulsion of sub-micron particles is often overwhelmed by Brownian motion, which erases temporal correlations in particle orientation and position. The motion of colloids in 3D along prescribed trajectories requires propulsion mechanisms that compete favorably with both gravitational and thermal forces. In this context, alternating current (AC) electric fields provide an attractive route to power and orient particle motions by induced charge electrophoresis (ICEP) as directed by particle asymmetries[18–22]. While several studies have highlighted the role of shape[18,23–25] and patchiness[15] on directing particle motions, few experiments have shown how tailored asymmetries can guide complex motions in 3D[26–28].

Here, we show that asymmetric metal patches on the surface of spherical colloids can direct their motion along non-linear helical trajectories powered by AC-electric fields[24,29]. The details of particle motion such as the speed and radius of the trajectory can be tuned by changing the size and shape of the metal patch. Particles with triangular patches of approximate mirror symmetry perform helical motions of both right and left handedness, depending on their initial orientation in the field. We demonstrate that helical trajectories can provide a functional advantage to particles navigating through porous membranes as compared to linearly propelled particles. These results suggest strategies for designing microrobots capable of navigating structured environments such as those found in advanced materials, geological formations, and biological tissues[10,17].

## Results

**Particle fabrication and propulsion.** We use polystyrene microspheres with metallic patches as a model colloid, hereafter called metallodielectric particles. The metal patch is introduced on the surface of an isotropic particle by glancing angle deposition[30,31]. As detailed in the Methods, a concentrated dispersion of negatively charged microspheres is assembled into a close-packed monolayer on a glass substrate. The substrate is then transferred to a physical vapor deposition chamber, where a 10 nm layer of chromium followed by a 30 nm layer of gold is deposited onto the particles. The angle $\varphi$ between the particle-coated substrate and the incident metal vapor is varied in the range $0° < \varphi < 20°$ (Fig. 1a)[31]. At these small angles, the particles in the monolayer are partially shaded from the vapor beam by neighboring particles. This phenomenon leads to the formation of metal patches of tunable size and shape on the particle surfaces

(Fig. 1b, c, Supplementary Fig. 1)[30]. By aligning one axis of the crystalline monolayer parallel to the vapor beam, we create particles with triangular patches with approximate mirror symmetry (Fig. 1b, c). By varying the angle $\varphi$, we control the fraction of the particle surface covered by the metal patch, denoted $f = A_{patch}(4\pi a^2)^{-1}$, where $A_{patch}$ is the patch area and $a$ is the particle radius[31]. The metallodielectric particles are dispersed in deionized water and positioned between two coplanar electrodes (Fig. 1d). A high frequency AC-electric field (**E**) is applied, and the particle dynamics is monitored by an upright light microscope. Remarkably, we observe that particles with triangular metal patches perform steady helical motions along the axis of the field (Fig. 1e, f–i and Supplementary Movie 1), while no motion was observed for non-patchy particles (Supplementary Fig. 2). When the field is turned off, the helical motions cease, and the particles diffuse freely in the dispersing medium. By cycling the field on and off, we observe that the particles are capable of four distinct helical trajectories that differ in the direction of travel along the field axis (positive or negative $z$-direction) and in the chirality of the trajectory (right- or left-handed) as discussed later (see also Supplementary Fig. 3).

**Kinematics of helical particle propulsion.** Particle motions are well described by the following kinematic equations for a helical trajectory of radius $R$, angular velocity $\boldsymbol{\Omega}$, and linear velocity $\mathbf{U}$

$$\mathbf{x}(t) = R\sin\boldsymbol{\Omega}t \quad \mathbf{y}(t) = -R\cos\boldsymbol{\Omega}t \quad \mathbf{z}(t) = \mathbf{U}t \qquad (1)$$

The four trajectories observed in experiments correspond to the four permutations in the sign of the angular frequency and of the linear velocity, for example, $\boldsymbol{\Omega} > 0$ and $\mathbf{U} > 0$ describe particle motion along a right-handed helix in the positive $z$-direction. We track the particle position in the $xz$-plane as a function of time and fit the kinematic model (1) to the data to estimate the parameters $R$, $\boldsymbol{\Omega}$, and $\mathbf{U}$ (Fig. 2a, b). Changing the magnitude of the applied field $E$ influences the speed of the particle motion but not the geometry of its helical trajectory. In particular, both the angular frequency $\Omega$ and the linear speed $U$ increase as the square of the field strength, while the radius $R$ and pitch $2\pi U\Omega^{-1}$ of the helix are independent of $E$ (Fig. 2c, d).

To understand how a particle selects its direction of motion along the field axis and the handedness of its helical trajectory, we toggle the AC-electric field on and off repeatedly and observe the resulting particle motion (Fig. 3a, b, Supplementary Movie 2). We find that the helix radius remains nearly constant for each field on-off cycle (Supplementary Fig. 5); small variations in the radius are likely caused by electrokinetic flows within the chamber and/or thermal motion of the particles (discussed further below). The probability that a particle reverses the handedness of its helical path depends on the time interval $\Delta t$ that the field is switched off. Experimentally, we count the number of handedness reversals for 100 on-off cycles of a prescribed duration $\Delta t$. The probability of handedness reversal, $P$ is estimated from the observed frequency as a function of $\Delta t$ for four particles of different sizes (Fig. 3c). This reversal probability is small for short times and approaches 0.5 for longer times, at which right- or left-handed helices are selected with equal probability. The characteristic reversal time for which $P(\Delta t) \approx 0.5$ (i.e., equiprobability) increases as the cube of the particle radius $a$ in quantitative agreement with the rotational diffusion time for a sphere, $D_r^{-1} = 8\pi\eta a^3(k_BT)^{-1}$, where $\eta$ is the viscosity of fluid, $k_B$ is the Boltzmann constant, and $T$ is temperature (Fig. 3d). These observations suggest that the selection of the particle trajectory is determined by the initial particle orientation when the field is applied.

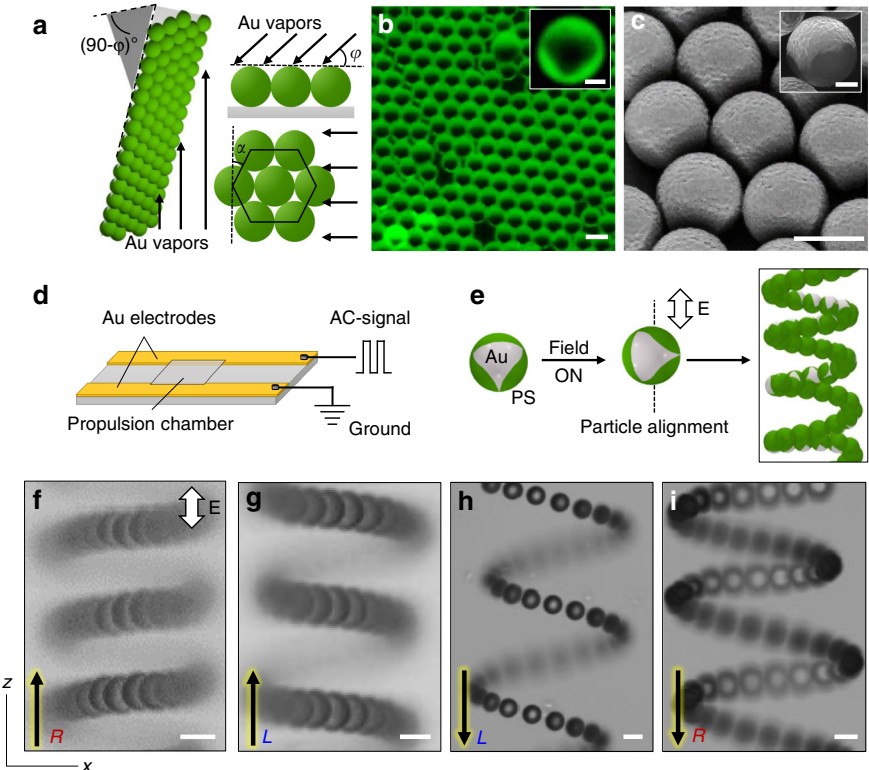

**Fig. 1** Particle fabrication and principle of directing dynamics of patchy colloids. **a** A schematic of glancing angle deposition of metal (gold, Au) vapors onto a substrate coated with monolayer of polystyrene (PS) microspheres. The arrows represent the direction of incident metal vapors. The shape of the metal patch depends on the orientation of the deposition vector relative to the particle monolayer as characterized by the angles $\varphi$ and $\alpha$. Triangular patches with (approximate) mirror symmetry are obtained when $\alpha = 0$. **b** Fluorescence and **c** scanning electron micrographs of polystyrene microspheres (green in **b** and dark gray in **c**) with a triangular shaped gold patch (dark in **b** and light gray in **c**). **d** A schematic of the experimental setup with coplanar gold electrodes. Particle motion in the AC-electric field is monitored by an upright light microscope in the direction normal to the plane containing the electrodes. **e** Schematic showing the initial alignment of the particle upon turning on the external AC-electric field. The particle aligns with in the applied field **E** (Supplementary Fig. 4) and performs helical motion. **f–i** Superimposed bright-field micrographs of patchy particles migrating along the field axis in four distinct helical trajectories. The applied field strength in the experiments is 250 V cm$^{-1}$ at a frequency of 10 kHz. The arrows represent the direction of linear motion of the helically propelled patchy colloids; $R$ and $L$ denote right and left-handed helical trajectories. The scale bars in **b**, **c**, and **f–i** are 5 μm, and in inset **b**, **c** are 1 μm

**Origin of helical motion of patchy particles**. Application of an AC-electric field to the aqueous dispersion induces an asymmetric fluid-flow around each metallodielectric particle that propels it though the fluid. This phenomenon, known as induced charge electrophoresis (ICEP)[20,29,32–35], is caused by the action of the external field on that portion of the ionic charge within the double layer induced by the field itself. For a particle moving in an unbounded fluid subject to an electric-field **E**, the translational and rotational velocity of the particle are predicted to have the following forms

$$\mathbf{U} = \frac{\varepsilon a}{\eta}\mathbf{C} : \mathbf{EE} \quad \text{and} \quad \mathbf{\Omega} = \frac{\varepsilon}{\eta}\mathbf{D} : \mathbf{EE} \qquad (2)$$

where $\varepsilon$ is the permittivity of the electrolyte, and **C** and **D** are dimensionless tensors[19]. Importantly, these tensors share the symmetry of the particle and are uniquely specified by its shape and composition[24]. As detailed in Supplementary Note 1, particles with a single plane of mirror symmetry ($C_S$ point group) are capable of three qualitatively distinct dynamical behaviors, one of which corresponds to the helical motions observed in experiment.

Analysis of the model suggests that particles with triangular metal-patches align their plane of mirror symmetry neither parallel nor perpendicular to the field axis but rather at an oblique angle $\theta$ (Fig. 4a, Supplementary Fig. 7). With this angle fixed, the particles rotate about the field axis at a steady rate **Ω** and translate with a velocity **U**, which is constant in the particle frame of reference. These predictions imply that particles trace helical paths along the field axis in qualitative agreement with the experimental observations. For any such path, there exist four equivalent trajectories obtained by reflection about an axis parallel and/or perpendicular to the field axis (Fig. 4b, Supplementary Fig. 8). Depending on its initial orientation, the particle selects one of these four trajectories upon application of the field. Moreover, the dynamical equations (2) imply that the speed of particle motion increases as the square of the field strength without altering the shape of the helical path (Fig. 2c, d).

Symmetry arguments alone cannot predict the details of particle motion such as the radius and pitch of the helical trajectory, which are specified implicitly by the tensors **C** and **D**. These quantities are expected to depend on the precise geometry of the triangular patch. Indeed, changing the size of the patch by altering the angle $\varphi$ of vapor deposition results in helical trajectories of different radii $R$ (Fig. 4g). By increasing the fractional patch area $f$ from 0.002 to 0.09, the helix radius increases more than two-fold from 10 to 22 μm. By solving the relevant electrostatic and hydrodynamic fields surrounding a particle with a prescribed patch shape, it should be possible to predict and ultimately design the motion of particles along trajectories of increasing complexity[24]. Preliminary attempts in developing such predictive models suggest that the details of the

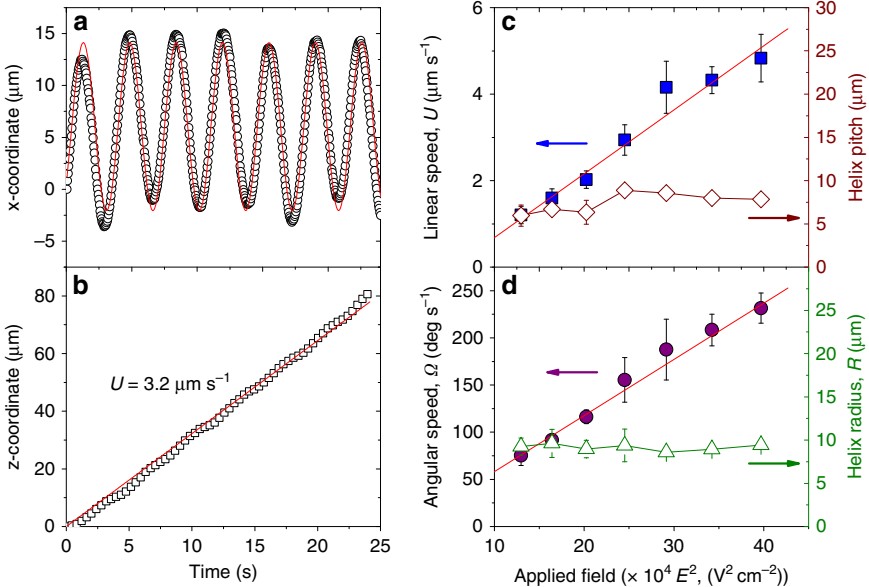

**Fig. 2** Characterization of the helical trajectory of patchy particles in electric field. **a, b** x- and z- location of a metallodielectric particle self-propelling along a helical path in electric field (250 V cm$^{-1}$ and 10 kHz). The scattered points represent the experimental data, and lines are fit using equation (1). The data-fitting provides the angular speed, helix radius, and linear speed in the z-direction. **c, d** Increase in the linear and angular speed of the particle performing helical motion with increasing strength of the applied electric field. The velocities scale linearly with $E^2$, which agrees with predictions based on ICEP. The pitch and radius of the helical trajectory are independent of the field strength. The data are obtained on one specific particle of radius 1.4 μm with metal patch fabricated using deposition angle $\varphi = 10°$. The error bars in **c** and **d** are standard deviation of the corresponding data set at least ten helical rotations

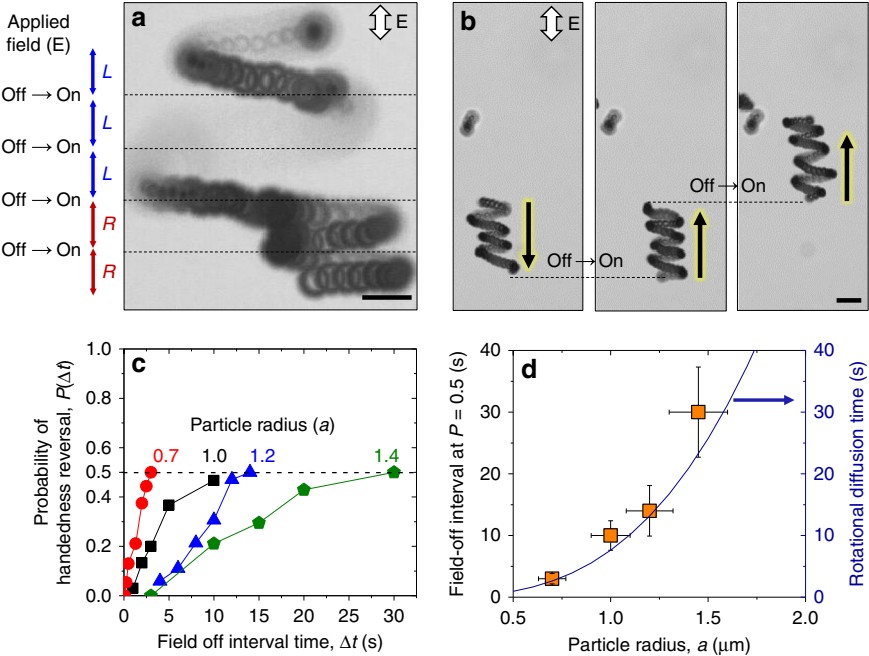

**Fig. 3** Dynamic switching of handedness and direction of propulsion. **a, b** Microscope images showing reversals of handedness and direction of helical trajectories when cycling the electric field on and off for particles of radius 1.2 μm and $\varphi = 5°$. The applied field strength is 250 V cm$^{-1}$ at 10 kHz, and field-off interval is 5 s for **a** and 10 s for **b**. The scale bars in **a** and **b** are 5 μm. **c** Experimentally determined probability for reversing the handedness of helical trajectories as a function of the field-off interval duration, $\Delta t$. Along the dashed line, the particle is equally likely to adopt helical motions of either handedness. The numbers in the plot represent the particle radius (in μm) used in determining the probabilities $P(\Delta t)$ of corresponding color. **d** Increase in characteristic field-off interval for equiprobability of inverting helix chirality with increasing particle radius. Here, the points represent experimentally determined data and the solid line corresponds to the theoretically calculated diffusion time. The y-error bars in **d** refer to the standard deviation of probability values for multiple experiments and x-error bars refer to the polydispersity of the particle size

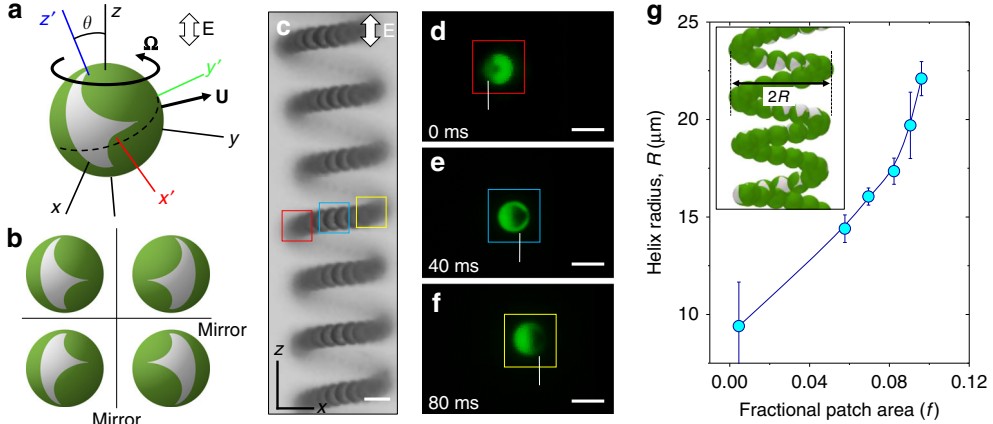

**Fig. 4** Patch orientation and effect of its size on helical motion. **a** Schematic showing the reference frames and velocity (angular and linear) vectors of a particle aligned at an oblique angle ($\theta$) with the axis of the applied field (z-axis). **b** Schematics of the four degenerate states of a particle with a triangular patch in uniform AC-electric field. The four stable orientations result in the four distinct helical trajectories of the particle. **c** Superimposed bright-field micrographs of a patchy particle migrating along cylindrical helical trajectory in an AC-electric field of strength 250 V cm$^{-1}$ and frequency 10 kHz. **d–f** Fluorescence micrographs of the particle with a triangular metal patch performing helical motion. Here the color of the boxes corresponds to the location of the particle along its helical path as shown in c (Supplementary Movie 3). Scale bars shown in **c–f** are 5 µm. **g** Increase in the radius of the helical trajectory with increasing size of the triangular metal patch at constant electric field of 325 V cm$^{-1}$ and frequency 10 kHz and particle radius $a = 2.6$ µm. The error bars in **g** refer to the standard deviation of the data set of at least ten measurements

particle's electric double layer can significantly impact the qualitative features of its motion in the field (Supplementary Notes 2 and 3). Using a plausible model of the double layer, numerical solutions of the electrokinetic equations predict that particles with triangular patches trace circular orbits normal to the applied field (Supplementary Fig. 9). The predicted speed of particle motion, the orbit radius, and its dependence on the patch size agree qualitatively with experimental observations (Supplementary Fig. 10); however, helical motions are not reproduced by this model under the investigated conditions.

## Discussion

In addition to ICEP, the observed helical motions of patchy particles may be influenced by several secondary factors such as (1) deviations from mirror symmetry in the patch geometry, (2) spatial gradients in the applied field, (3) gravitational forces and torques, (4) electro-osmotic flows within the chamber, (5) transient effects during particle polarization, (6) interactions with system boundaries, and (7) Brownian motion. The fabricated particles with triangular patches do not have perfect mirror symmetry and are therefore chiral; however, such imperfections cannot be responsible for producing both right- and left-handed helices in a single particle (Fig. 3a). Spatial gradients in the applied field can drive particle motion due to dielectrophoretic forces; however, these effects are predicted to be negligible in the center of the chamber where helical motions are observed (see Supplementary Note 4, Supplementary Fig. 11). Gravitational forces and torques can lead to translation and rotation of patchy particles; however, these motions are predicted to be small compared to the observed ICEP motions (see Supplementary Note 5). Field-induced fluid flows along the boundaries of the chamber are not expected at the applied frequencies ($\omega \gg \kappa D L^{-1}$, where $D$ is the ion diffusivity, $\kappa^{-1}$ is the screening length, and $L$ is the width of the chamber) as confirmed by experiments on passive (non-patchy) particles (Supplementary Fig. 2)[20]. At the same time, the applied frequency may be comparable to the particle charging time[20] ($\omega \sim \kappa D a^{-1}$), such that transient effects in double-layer polarization (neglected in our model) may contribute to ICEP motions. Also absent from the present model are effects due to electrostatic and hydrodynamic interactions with

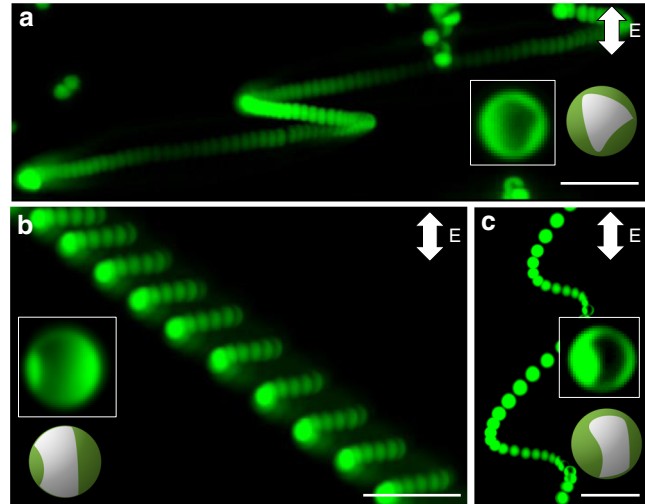

**Fig. 5** Role of patch shape in directing particle trajectory. **a–c** Superimposed fluorescence images of particles performing motion along non-cylindrical helical trajectories in applied electric field (Supplementary Movie 4). The insets are the fluorescence micrographs and schematics of the particles showing asymmetric metal-patch shapes (dark regions). The experiments were performed in AC-electric field of strength 250 V cm$^{-1}$ and frequency 10 kHz. The unusual particle trajectories derive from the low-symmetry metal-patches shown in the insets. Scale bars in **a–c** are 25 µm

the walls of the chamber as well as Brownian motion of the particles (see Supplemental Movies 1–3). Further work is required to more fully elucidate the origins of the observed helical motions and the influence of these many secondary factors.

By further reducing the symmetry of the metal patches, we observe particle motions of increasing complexity as shown in Fig. 5a–c. In these experiments, particles with low-symmetry patches are prepared by varying the relative orientation of the colloidal crystal monolayer with respect to the incident vapor beam (i.e., by varying the angle α in Fig. 1a). Fixing the polar angle ($\varphi = 10°$), rotation of the crystal monolayer can transform the triangular patches (α = 0°) described above into curved

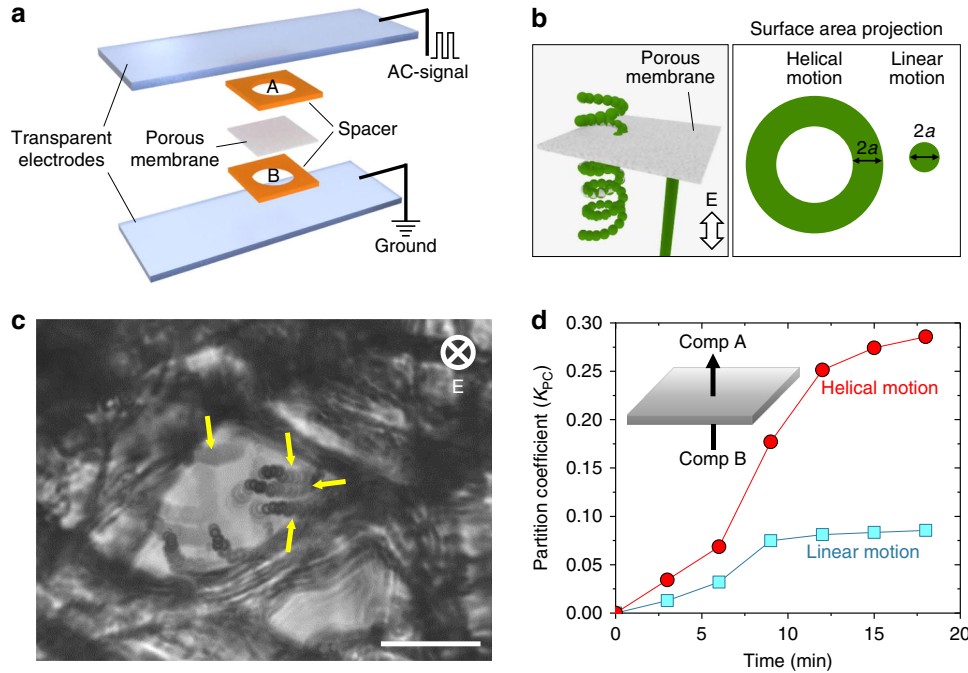

**Fig. 6** Unique ability of helically propelled active colloids to tunnel through crowded matrices. **a** Schematic of the experimental setup used for particle tunneling experiments. Initially the patchy particles were confined only into compartment B, and their tunneling into compartment A through a porous membrane was monitored and quantified using a light microscope. The AC-electric field was applied normal to the plane of porous membrane. **b** Schematics representing the helical and linear propulsion of particles through the membrane, and projected area of the particle trajectories on the membrane. **c** Microscope image showing the tunneling of several helically propelled particles through a crosslinked cellulose matrix. Here $E = 260 \, \text{V cm}^{-1}$ at 10 kHz, $a = 1 \, \mu\text{m}$, and $\varphi = 10°$. The arrows indicate the particle trajectories. **d** Increase in the fraction of particle penetrating through the porous membrane i.e., partition coefficient with time. The circles represent the helically propelling patchy particles and squared are linearly propelled particles. Scale bar in **c** is 20 μm

rectangular patches for $\alpha = 30°$[30,36]. Particles with such asymmetric patches are observed to move along skewed helical trajectories on axes oblique to that of the applied field (Fig. 5a–c).

Complex particle trajectories may be useful in enhancing the transport of microrobots through structured media and crowded environments. To demonstrate this possibility, we investigate how the motion of microparticles along helical trajectories can influence their transport through a crosslinked cellulose membrane (Fig. 6, Supplementary Movie 5). A chamber between two parallel electrodes is divided into two compartments A and B by a porous membrane of thickness 210 μm with disordered irregular pores of sizes in the range 20–45 μm (Fig. 6a, b). Particles are introduced into the bottom compartment B and migrate through the membrane upon application of an electric field directed normal to the membrane (Fig. 6b). The transient distribution of particles within the chambers is quantified by a partition coefficient, $K_{PC}(t) = n_A(t) n_{tot}^{-1}$, where $n_A(t)$ is the number of particles in compartment A at time $t$, and $n_{tot}$ is the total number of particles. The number $n_{tot}$ is determined experimentally by fluorescence imaging of the bottom plane of compartment B containing all the particles at time $t = 0$ min (typically, $n_{tot} \sim 800$ particles). The number $n_A(t)$ is estimated by changing the focal plane to the-bottom of compartment A and monitoring the increase in particle number with time. The corresponding coefficient $K_{PC}(t)$ increases from zero at the start of the experiment (since $n_A(0) = 0$) as particles tunnel from compartment B to A.

We compare the behavior of particles moving along helical trajectories to those moving along linear trajectories normal to the membrane. First, we consider metallodielectric particles with triangular patches moving by ICEP along helical trajectories in an AC-electric field. For comparison, we consider negatively charged polystyrene spheres (zeta potential of −40 mV) moving by

electrophoresis along linear trajectories in a direct current (DC)-electric field directed from chamber A to B (Fig. 6a, b). A DC field strength of $E = 2 \, \text{V cm}^{-1}$ is selected to drive motion with a linear speed of 5 μm s$^{-1}$, comparable to that of helical ICEP motions. For both particle types, the partition coefficient $K_{PC}(t)$ increases in time, approaching a constant value after ca. 15 min (Fig. 6d). Importantly, both the initial rate of increase and the asymptotic partition coefficient is larger (by 2–3 times) for the helical trajectories than for the linear trajectories.

One plausible explanation for the enhanced tunneling of particles with helical dynamics is the increase in the sampling area due to the rotational component of particle motion (Fig. 6b). To see this, we consider an idealized membrane formed by a planar wall with a single pore through which the particle can pass. When a linearly propelled sphere of radius $a$ collides normal to the wall, it samples a relatively small area $\pi a^2$ and stops. By contrast a helically propelled particle continues to move along a circular orbit parallel to the wall thereby sampling a larger area of $\sim 4\pi a R$ and increasing its chances of finding the pore (Fig. 6b). In this way, rotational motions may help the particle to 'search' for passages across the porous matrix. Further evidence for this hypothesis is provided by tunneling experiments on different types of patchy particles that perform helical motions of different radii $R$. We observed that the partition coefficient after 15 min increases monotonically with the helix radius $R$ (Supplementary Fig. 6). These experiments highlight the potential value of active particles capable of complex trajectories, which might be optimized for navigating specific environments.

## Conclusions

In conclusion, we presented a new method of encoding 3D helical motion in self-propelling colloids by programing the force and

torque distribution on the particle surface. A combination of particle, metal-patch, and applied field characteristics allows to predefine the particle trajectory. In addition, we demonstrated the particles to navigate through complex crosslinked matrices, which may facilitate in designing next generation microrobots for advanced biomedical applications. The study lays a foundation to better understand the impact of surface force distribution on the kinematics of active particles and presents a new principle of programming dynamics of artificial colloidal matter.

## Methods

**Patchy particle fabrication.** Aqueous suspensions of surfactant-free polystyrene latex spheres ($a = 0.7, 1.0, 1.2, 1.4$, and $2.6 \mu m$) were purchased from Magsphere Inc. (CA) and Molecular Probes (OR). The particles were pre-functionalized with negatively charged carboxylate groups with a measured zeta potential of $-40$ mV (Anton Paar, Litesizer 500) at pH 7. Prior to use, the particles were washed three times with deionized water (DI water) and concentrated to 10% by weight. Particles monolayers were deposited onto clean glass slides by the convective assembly method[30] and coated by a 10 nm chromium layer (deposition rate: $0.5 \text{ nm s}^{-1}$) followed by a 30 nm gold layer (deposition rate: $0.10 \text{ nm s}^{-1}$) under vacuum ($1 \times 10^{-6}$ Torr) in a custom-built thermal evaporator. The metal-coated particles were transferred into a plastic vial by a spatula and re-dispersed in DI water (concentration ~0.01 wt %) by gentle sonication before the experiments.

**Experimental setup.** The electrodes were prepared by gold deposition on microscope glass slides ($3 \times 1$ inch, VWR) with a 2 mm gap. The electrodes were soaked in Nochromix solution for 24 h and rinsed thoroughly with DI water before the experiments. Teflon masking tape was used to form a $2 \text{ mm} \times 2 \text{ mm} \times 100 \mu m$ chamber between the gold electrodes. The patchy particle suspension was transferred to the chamber and capped by a cover glass to create a Hele-Shaw cell type configuration. The tunneling experiments shown in Fig. 6 were performed using two indium tin oxide coated glass electrodes (resistance: $8–12 \Omega \text{ sq}^{-1}$) arranged parallel to one another and perpendicular to the direction of gravity (Fig. 6a). The thickness of cellulose membrane used in the study was 210 $\mu m$ with pore diameter in the range 20–45 $\mu m$. The pores in the membrane were asymmetric and randomly distributed. The membrane was suspended between the electrodes by two 0.5 mm thick silicone spacers (Sigma–Aldrich). Initially, the particles were confined to the lower half of the chamber and moved upward against gravity upon application of the field. In both experimental configurations, the electrodes were connected to a function generator (Agilent) and a high-voltage amplifier (Tegam). A square wave AC-electric field of magnitude $130–650 \text{ V cm}^{-1}$ and frequency 4–15 kHz was applied to the dispersion and monitored by an oscilloscope (Keysight).

**Imaging and microscopy.** To characterize the patch geometry, the metallodielectric particles were sputter coated with platinum and imaged by backscattered scanning electron microscopy (JEOL, Model JSM-6610 LV). Particle dynamics in the applied field were captured by a Leica DM6 B optical microscope equipped with a Leica DFC9000 GTC camera.

**Particle tracking.** Particle motions were analyzed using the ImageJ software[37]. For a given video, each frame was converted to a binary image that distinguished the particle from its surroundings. For each frame, the particle position ($x, z$) in the imaging plane was then computed as the centroid of the particle region using the "Centroid" feature in ImageJ's "Analyze Particles" menu. The helix radius, angular velocity, and linear velocity were estimated from these data by least squares regression using the kinematic model (1).

## Data availability

The data that support the findings of this study are available from the corresponding author on request.

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

## Acknowledgements
The authors thank Prof. K. McPeak (LSU) and Sara K. F. Stofela (LSU) for assistance with patchy particle synthesis, Ms. Stacey Wieseneck (LSU) with ICEP experiments, and Profs. O.D. Velev (NCSU), V.T. John (Tulane), M. Hjortsø (LSU), and M. Schoen (TU Berlin) for useful discussions. The synthesis of particles, experiments on their self-propulsion, and their interpretation by J.G.L, W.A.S., and B.B. was supported by U.S. Department of Energy, Office of Science, Basic Energy Sciences, under EPSCoR Grant No. DE-SC0012432 with additional support from the Louisiana Board of Regents. The theoretical analysis performed by A.M.B. was supported in part by the National Science Foundation Graduate Research Fellowship Program under Grant DGE1255832. The modeling of particle propulsion developed by K.J.M.B. was supported in part by the Center for Bio-Inspired Energy Science, an Energy Frontier Research Center funded by the US Department of Energy, Office of Science, Basic Energy Sciences under Award DE-SC0000989.

## Author contributions
B.B. conceived the project and designed the experiments. J.G.L. performed the experiments and analyzed the data following the directions from B.B. and K.J.M.B. K.J.M.B. and A.M.B. developed the mathematical model for helical propulsion. B.B. and W.A.S. developed the physical basis of enhanced particle tunneling through membrane. All authors participated in writing the manuscript.

## Additional information

**Competing Interests:** The authors declare no competing interests.

