## [Peer Review File · Nature Communications]

Reviewers' comments:

Reviewer #2 (Remarks to the Author):

Lee et al 'Directed propulsion of spherical particles along 3D helical trajectories' show that patchy metal-dielectric particles can move in 3d helical trajectories in an ac electric field. The paper is to some extent a validation of a colloidal trajectory discussed in a theory paper by two of the authors (Proc. Natl. Acad. Sci. U.S.A. 115, E1090-E1099, 2018). In addition to the helical trajectories, which are beautifully shown, the paper discusses the penetration of colloids through a porous membrane. This is presented as an application and it is argued that helical trajectories facilitate the motion through the membrane. If this is the case, then this is an interesting observation, but it is not clear how general the experimental results and observations are. In my opinion, further experiments are required if the authors want to show that helical trajectories help with transport through porous media.

1 The membrane experiments do not seem to be systematic. The authors should discuss how particle size and type affects the results. The authors should vary the trajectories systematically and show how this correlates with membrane penetration. What exactly is the pore structure in the membranes and how does transport correlate with the particle size and trajectories? In its present form the results the experiments seem incomplete and too qualitative to be useful.

2 In the porous membrane penetration experiments it remains unclear how the number of particles in the compartment A (and also the total number of particles) is counted. How do the authors ensure that every particle in compartment A is considered in the statistics? A detailed explanation of this procedure should be given in the experimental section. In this context, the authors should also provide the total number of particles used in the experiments in Fig. 4d.

3 In Figure 3g, it is reported that the helix radius depends on the fractional patch area. But in Figure 2e the helix radius can be seen to change randomly for the same particle between the different electric field cycles.

4 Does the bottom heaviness (e.g. changing with fractional patch area) play a role in the dynamics in these experiments?

5 The authors describe the probability of switching the chirality of motion based on the "off" times of the electric field in Fig 2g. The authors should mention how many particles are considered to arrive at the probability values.

6 The theoretical model for Fig.1 assumes that the electric field is in the plane of the colloids. How thick are the electrodes? Is the field homogeneous throughout the in which the colloids move. Can the authors discuss how any in-homogeneity, if it is present, affects the trajectories?

7 Could the authors report a control experiment on motion of passive tracers (e.g. negatively charged polystyrene spheres, as used in the top-bottom electrode configuration in Figure 4), for the in-plane electrode configuration in Fig 1.

8 The authors provide experiments and theory of the particle motion. Although the exact form of the tensors C and D are dependent on the metal-patch geometry it would be nice to calculate an expectation value of the particles velocities (e.g. for the movies in video 1 or Fig. 2b) for the given parameters and compare it to the result of the measurements to prove the theory is correct.

9 The meaning of the arrow in Fig. 2 g is unclear.

10 Line 222 mentions Fig. 5c) which is not existent?

Reviewer #3 (Remarks to the Author):

The manuscript describes induced-charge electrophoretic motion of a colloid with a patchy metal-dielectric structure. The trajectory of such a colloid driven by an AC electric field is found to be a 3D helix. The work is potentially of interest but the interpretation and presentation of the results are not convincing.

1. The biggest problem is in interpretation of the results, namely, in the following two statements that represent the main result of the work:

(i) that the chirality of motion is set by the initial orientation of colloids in the field and

(ii) that the colloids are achiral. I see no proof of either statement. T

The deposited region of metal might well lead to chiral structure. It is not clear why a colloid with a metal patch might have different orientation in the field. Once the field is applied, the colloid should align somehow; this alignment depends on the geometry of the metal island and most likely is unique.

Some major questions are left without answers:

2. Why the radius of trajectory varies with the patch area?

3. Why the pitch and radius of the trajectory are field-independent?

4. Why the linear velocity increases with the field?

Less serious issues:

5. Figure 1a is not clear; the geometry of metal deposition should be presented in a better way, with both polar and azimuthal angles clearly depicted.

6. Introductory part claims that micron-size particles are significantly influenced by gravity.

Apparently, the authors means particles larger than one micrometer.

7. The references 14,15 on line 38 to ICEP should be supplemented by references to theoretical works (Bazant, Squires, others) where the concept has been introduced.

8. The argument about better permeability of helicoidally transported particles is questionable: For a real membrane with non-flat local profile, a linear particle can also explore different regions of the membrane, following downhill slopes.

Reviewer 2

General comment: Lee et al 'Directed propulsion of spherical particles along 3D helical trajectories' show that patchy metal-dielectric particles can move in 3d helical trajectories in an ac electric field. The paper is to some extent a validation of a colloidal trajectory discussed in a theory paper by two of the authors (Proc. Natl. Acad. Sci. U.S.A. 115, E1090-E1099, 2018). In addition to the helical trajectories, which are beautifully shown, the paper discusses the penetration of colloids through a porous membrane. This is presented as an application and it is argued that helical trajectories facilitate the motion through the membrane. If this is the case, then this is an interesting observation, but it is not clear how general the experimental results and observations are. In my opinion, further experiments are required if the authors want to show that helical trajectories help with transport through porous media.

Response: We thank the Reviewer for the positive evaluation of our work and insightful comments which helped us to improve the manuscript further.

Comment 1: The membrane experiments do not seem to be systematic. The authors should discuss how particle size and type affects the results. The authors should vary the trajectories systematically and show how this correlates with membrane penetration. What exactly is the pore structure in the membranes and how does transport correlate with the particle size and trajectories? In its present form the results the experiments seem incomplete and too qualitative to be useful.

Response: The porous membrane used is a commercially available cellulose filter with irregularly shaped, disordered pores with a mean pore size of 20-45 μm . We have clarified this in the revised manuscript. We have also performed additional experiments where we systematically increase the radius of the helical trajectory (by changing the patch size) and monitor the corresponding change in the membrane penetration. The result is shown in Figure S6 and discussed in the revised manuscript text (given below). Consistent with our simple geometric arguments, we find an increase in the membrane penetration with increasing radius of the helical trajectory (Fig. 4b). We have also revised Supplemental Movie S5 that shows how a particle of smaller helix radius is trapped in the membrane, while particles with larger helix radius more effectively scans the membrane in "search" of a pore.

We agree with the reviewer that particle size would also affect the membrane penetration. However, we have not explored the effect of particle size due to the fact that upon increasing the size (above 3 μm) significant gravitation force is operational on the particles affecting the self-propulsion, and upon lowering the particle size (below 1 μm) Brownian diffusion impacts the trajectory. Further experiments of active penetration in density-matched solvent would be necessary to perform such studies, which are beyond the scope of the present work.

Added text in the manuscript on page 10:

"Further evidence for this hypothesis is provided by tunneling experiments on different types of patchy particles that perform helical motions of different radii R . We observed that the partition coefficient after 15 minutes increases monotonically with the helix radius R (Fig. S6)."

Comment 2: In the porous membrane penetration experiments it remains unclear how the number of particles in the compartment A (and also the total number of particles) is counted. How do the authors ensure that every particle in compartment A is considered in the statistics? A detailed explanation of this procedure should be given in the experimental section. In this context, the authors should also provide the total number of particles used in the experiments in Fig. 4d.

Response: The two compartments in the experimental setup of Fig. 4 lie in different focal planes. Fluorescence microscopy therefore enables us to distinguish the particles within the two compartments. We estimated the number of particles in each compartment by counting particles in fluorescence images focused on the upper and lower compartments. We provide further details in the manuscript by including the following sentences on page 10:

“The number n_{tot} is determined experimentally by fluorescence imaging of the bottom plane of compartment B containing all the particles at time $t = 0$ min (typically, $n_{tot} \sim 800$ particles). The number $n_A(t)$ is estimated by changing the focal plane to the bottom of compartment A and monitoring the increase in particle number with time. The corresponding coefficient $\kappa(t)$ increases from zero at the start of the experiment (since $n_A(0) = 0$) as particles tunnel from compartment B to A.”

Comment 3: In Figure 3g, it is reported that the helix radius depends on the fractional patch area. But in Figure 2e the helix radius can be seen to change randomly for the same particle between the different electric field cycles.

Response: The apparent change in helix radius in Fig. 3g is an illusion due to the change in handedness combined with the visualization of the trajectory in a 2D plane. The Field ON interval is less than the characteristic time-period of helical motion, which leads to the illusion of change in radii in the overlaid 2D images. We quantify the helix radius by fitting the individual particle paths in the field ON state as shown Figure S5. The figure demonstrates that the helix radius remains nearly constant for the field on off experiments. Small differences ($\sim 10\%$) in the helix radius values estimated can be attributed to variety of secondary factors, which are discussed in more detail within the revised manuscript on Page 9. We highlight this aspect on page 5 by including following text:

“We find that the helix radius remains nearly constant for each field on-off cycle (Fig. S5); small variations in the radius are likely caused by electrokinetic flows within the chamber and/or thermal motion of the particles (discussed further below). The probability that a particle reverses the handedness of its helical path depends on the time interval Δt that the field is switched off.”

Comment 4: Does the bottom heaviness (e.g. changing with fractional patch area) play a role in the dynamics in these experiments?

Response: For the experimental conditions, we estimate that gravitational effects are small relative to ICEP motions. These estimates are detailed in Supplementary Note 4 and referenced in the main text.

Comment 5: The authors describe the probability of switching the chirality of motion based on the "off" times of the electric field in Fig 2g. The authors should mention how many particles are considered to arrive at the probability values.

Response: Each probability estimate was determined by 100 on-off cycles in the applied field field, i.e. each point in Figure 2g corresponds to a set of 100 experiments. This information is provided in the following sentence on page 5:

“Experimentally, we count the number of handedness reversals for 100 on-off cycles of a prescribed duration Δt .”

Comment 6: The theoretical model for Fig.1 assumes that the electric field is in the plane of the colloids. How thick are the electrodes? Is the field homogeneous throughout the in which the colloids move. Can the authors discuss how any in-homogeneity, if it is present, affects the trajectories?

Response: The electrodes used in the experiment are 100 nm thick. The field is approximately homogeneous throughout most of the chamber except near the electrodes. This claim is supported by *Supplementary Note 4*, in which we compute the electric field in the chamber. In this article, we perform experiments in the center region of the chamber to avoid complications due to field gradients. Near the electrodes, field gradients exert dielectrophoretic (DEP) forces on the particle along the field direction, which result in helical motions of non-uniform pitch.

Comment 7: Could the authors report a control experiment on motion of passive tracers (e.g. negatively charged polystyrene spheres, as used in the top-bottom electrode configuration in Figure 4), for the in-plane electrode configuration in Fig 1.

Response: We have included the control experiments in Figure S2 in the revised supporting information. The figure shows lack of any significant motion of the non-patchy negatively charged polystyrene spheres (radius = 1 μm) in the co-planar electrode configuration.

Comment 8: The authors provide experiments and theory of the particle motion. Although the exact form of the tensors C and D are dependent on the metal-patch geometry it would be nice to calculate an expectation value of the particles velocities (e.g. for the movies in video 1 or Fig. 2b) for the given parameters and compare it to the result of the measurements to prove the theory is correct.

Response: Following the Reviewer’s suggestion, we developed a more detailed theory of ICEP motions for patchy spheres; however, we are yet to achieve quantitative agreement with the experimental observations. In *Supplemental Note 2*, we consider the case of thin double layers and show that helical motions are not possible in this limit. In *Supplemental Note 3*, we consider the more relevant case of finite double layers and describe numerical computation of the shape tensors C and D. The results depend sensitively on the equilibrium double layer surrounding the metallodielectric particles. For physically reasonable estimates of the surface charge and zeta potential, the model predicts circular motions normal to the field with speed and radius comparable to those observed in experiment. For the limited number of conditions investigated, we did not observe helical motions. Further work is required to explore the space of feasible model parameters (or alternative descriptions of the double layer) in search of such motions.

Comment 9: The meaning of the arrow in Fig. 2g is unclear.

Response: To avoid any confusion we have removed the arrow from the figure.

Comment 10: Line 222 mentions Fig. 5c) which is not existent?

Response: We have corrected the typo and replaced 5c with 4b.

Reviewer 3

General comment: The manuscript describes induced-charge electrophoretic motion of a colloid with a patchy metal-dielectric structure. The trajectory of such a colloid driven by an AC electric field is found to be a 3D helix. The work is potentially of interest but the interpretation and presentation of the results are not convincing.

Response: We thank the reviewer for acknowledging the potential of our work. Below we provide details of the changes introduced in the manuscript based on the reviewer's comments.

Comment 1: The biggest problem is in interpretation of the results, namely, in the following two statements that represent the main result of the work:

(i) that the chirality of motion is set by the initial orientation of colloids in the field and

(ii) that the colloids are achiral. I see no proof of either statement.

The deposited region of metal might well lead to chiral structure. It is not clear why a colloid with a metal patch might have different orientation in the field. Once the field is applied, the colloid should align somehow; this alignment depends on the geometry of the metal island and most likely is unique.

Response: First, we acknowledge that the particles (like any macroscopic objects) do not have perfect mirror symmetry. For the purpose of understanding their ICEP motion, it is instructive to consider the case of idealized particles with perfect mirror symmetry. As we show in Supplemental Note 1, such particles are capable of four types of helical motions in agreement with the experimental observations. If helical motion instead derives from particle chirality, we would not expect the same particle to perform helical motions of opposite handedness. We have modified the text on page 9 to clarify the approximate nature of the particle symmetry and to address the possible role of particle chirality in selecting the handedness of helical motions.

Second, we note that the alignment of a particle due to ICEP is fundamentally different from the alignment of a polarizable particle in an applied field (in the absence of field-induced fluid flows). For particles with mirror symmetry, a polarizable particle may align its symmetry plane parallel or perpendicular to the applied field but *never* oblique to the field. In ICEP, particle rotation is due both to electric torques and to field-induced fluid flows. As a result, a new possibility arises in which the particle can align its symmetry plane oblique to the field. This corresponds to 4 distinct stable orientations in the field as shown in Figure S8.

The experimental evidence presented in the paper strongly supports this physical picture. By switching the field on and off, a single particle is observed to perform four types of helical motion. To switch from one type of helical motion to another, the field must be off for sufficient time that the particle can change its orientation via rotational diffusion (Fig. 2e-h). We have now

included a new Figure S3 in supplementary information that shows the patch orientation of a single particle performing helical motions of opposite handedness. Additionally, Figure S4 shows how the particle orients upon application of the AC-electric field.

Some major questions are left without answers:

Comment 2: Why the radius of trajectory varies with the patch area?

Response: In general, the type and extent of particle asymmetry is expected to influence the shape of the particle's ICEP trajectory. The specific experimental trend shown in Figure 3g is reproduced qualitatively by the model detailed in *Supplementary Note 3* (see Figure S10). Physically, the radius of the helix is determined by the relative rates of translation and rotation. Apparently, decreasing the size of the patch (by changing the deposition angle) reduces the speed of translation more rapidly than it decreases the speed of rotation, thereby reducing the radius.

Comment 3: Why the pitch and radius of the trajectory are field-independent?

Response: The experimental observation that the pitch and the radius of the helical trajectory are independent of the applied field follows from equation (2) for the ICEP velocity. Scaling lengths by the particle radius a and time by the electroviscous time $\eta/\varepsilon E^2$, the particle dynamics depends only on the dimensionless shape tensors **C** and **D**, which are independent of the field strength. Consequently, the solution to these dynamics (i.e., the shape of the helix) is also independent of the field strength. This is provided on Page 7 of revised manuscript.

Comment 4: Why the linear velocity increases with the field?

Response: Again, this observation follows from equation (2) for the ICEP velocity. The field strength enters only through the time scale $\eta/\varepsilon E^2$. As a result, the speed of translation and rotation increase in proportion to the square of the field strength.

Less serious issues:

Comment 5: Figure 1a is not clear; the geometry of metal deposition should be presented in a better way, with both polar and azimuthal angles clearly depicted.

Response: We have updated Figure 1a with additional 2D schematic for better depiction of the polar (ϕ) and azimuthal angles (α).

Comment 6: Introductory part claims that micron-size particles are significantly influenced by gravity. Apparently, the authors means particles larger than one micrometer.

Response: For the present system, gravitational effects become significant for particles larger than few micrometers as discussed in a new *Supplementary Note 5*. The relative magnitude of gravitational and ICEP motions are influenced by several factors (not just size). We therefore prefer to keep the somewhat ambiguous "micron-sized" label, which is clarified later in the text.

Comment 7: The references 14,15 on line 38 to ICEP should be supplemented by references to theoretical works (Bazant, Squires, others) where the concept has been introduced.

Response: We have included additional references (Ref. 23, 24, 34, and 35 of the manuscript).

Comment 8: The argument about better permeability of helicoidally transported particles is questionable: For a real membrane with non-flat local profile, a linear particle can also explore different regions of the membrane, following downhill slopes.

Response: We agree that the geometric model we present is a drastic simplification of the actual transport process. However, additional experiments on the permeation of patchy particles with different helical motions provide further evidence for this picture. In particular, we show that the observed partition coefficient increases with increasing helix radius (Fig. S6). Further experimental and theoretical work is needed to investigate the quantitative details of particle transport through structured environments.

REVIEWERS' COMMENTS:

Reviewer #2 (Remarks to the Author):

The authors have addressed my comments and made changes to their manuscript. I am not entirely convinced that the helical trajectory will in general show enhanced transmission through membranes, especially as systemic tests of different sizes (Janus particles and membranes) are difficult to obtain, but the experiments are well described and they may stimulate further work. Nevertheless, I think the dynamics of the particles has been clearly elucidated and the helical paths of the generally achiral objects is fascinating and interesting and thus support the publication of this paper.

Reviewer #3 (Remarks to the Author):

The revised manuscript addresses my comments and I recommend it for publication.